# Nickel-catalyzed synthesis of 1,1-diborylalkanes from terminal alkenes

Lei Li[1], Tianjun Gong[1], Xi Lu[1], Bin Xiao[1] & Yao Fu[1]

Organoboron compounds play an irreplaceable role in synthetic chemistry and the related transformations based on the unique reactivity of C–B bond are potentially the most efficient methods for the synthesis of organic molecules. The synthetic importance of multiboron compounds in C–C bond formation and function transformation reactions is growing and the related borations of activated or nonactivated alkenes have been developed recently. However, introducing directly two boron moieties into the terminal sites of alkenes giving 1,1-diborylalkanes in a catalytic fashion has not been explored yet. Here we describe a synthetic strategy of 1,1-diborylalkanes via a Ni-catalyzed 1,1-diboration of readily available terminal alkenes. This methodology shows high level of chemoselectivity and regioselectivity and can be used to convert a large variety of terminal alkenes, such as vinylarenes, aliphatic alkenes and lower alkenes, to 1,1-diborylalkanes.

[1] Hefei National Laboratory for Physical Sciences at the Microscale, CAS Key Laboratory of Urban Pollutant Conversion, Anhui Province Key Laboratory of Biomass Clean Energy, iChEM, University of Science and Technology of China, Hefei 230026, China. Correspondence and requests for materials should be addressed to B.X. (email: binxiao@ustc.edu.cn) or to Y.F. (email: fuyao@ustc.edu.cn)

Organoboron compounds are recognized as versatile building blocks and fundamental intermediates in organic synthesis[1–5]. Particularly because these compounds are usually air and moisture stable and easy to handle compared with Grignard reagents, organolithium reagents, and also enable further transformations based on the unique reactivity of the C–B bond[6]. Transition-metal catalyzed conversion of sp[2]-hybridized carbon in alkenes to C(sp[3])-B bond for the formation of organoboron building blocks is a more recently developed class[7] (Fig. 1a). Several pioneer synthetic methods of momoborylalkanes, 1,2-diborylalkanes and 1,1,1-triborylalkanes from alkenes have been reported recently. Rh-catalyzed synthesis of 1,2-diborylalkanes from terminal vinylarenes was firstly reported by Westcott and co-workers, although the 1,2-diboration product was obtained in very low yield[8]. The enantioselective 1,2-diboration of terminal alkenes catalyzed by platinum catalyst, combined with Suzuki coupling reactions was achieved by Morken's group[9]. Huang's group also reported a Co-catalyzed protocol for the formation of 1,1,1-tri-borylalkanes, but the alkene substrates are restricted to vinylarenes[10].

As an important class of organoboron compounds, it is undeniable that the 1,1-diborylalkanes show significant applications in organic synthesis. They can either be manipulated in enantioselective catalytic fashion[11–15] (Fig. 1b) and or they can provide powerful synthetic module for concise synthesis of complex molecules through multiple C–C bond formation[16–22] (Fig. 1c), function transformation reactions (Fig. 1d), or both. However, direct synthesis of 1,1-diborylalkanes from terminal alkenes has not been achieved yet. If successful, both synthesis of 1,1-diborylalkanes in synthetic chemistry and applications of alkenes from petrochemical industry will be greatly improved.

## Results

**Reaction discovery.** As the major research focus of our group, efforts were devoted to discover new C–B bond forming reactions and alkenes functionalization reactions[55–57]. We now report the discovery of a synthetic method of 1,1-diborylalkanes, namely, Ni-catalyzed 1,1-diboration of terminal alkenes (Fig. 1e). This reaction provides a convenient strategy for the synthesis of 1,1-diborylalkanes from more stable and less expensive substrates (alkenes and boronic ester). The reaction shows high levels of chemo- and regio-selectivity. As reactions related to organoboron compounds have enjoyed great success in modern organic synthesis, the present reaction is expected to find important applications in synthetic chemistry.

**Investigation of reaction conditions.** Various bases, ligands, solvents and Ni catalysts for the 1,1-diboration reaction of 4-phenyl-1-butene 1 with bis(pinacolato)diboron (B$_2$pin$_2$) were screened (Table 1 and Supplementary Tables 1–4). Initially, we tried tricyclohexyl phosphine, a monodentate phosphine ligand that worked well for boration reactions. To our delight, the desired product 2 was obtained in 39% GC yield and only trace amount of 1,2-diboration byproduct 2′ was detected. Some other bases except LiOMe have weakened the 1,1-diboration and only Cs$_2$CO$_3$ showed a little effect giving 2 in 22% yield (entry 2–5). We then moved our attention to a sample of 1-analogs and nitrogen heterocyclic carbene (NHC) ligands. To our disappointment, these ligands were found to be ineffective and only trace amount of 2 and 2′ were detected (entries 6–8). We were enlightened by the optimization results that the ligands bearing dicyclohexylphosphine moiety may be good for the 1,1-diboration reaction. Surprisingly, when Cy-XantPhos was used, 2 was increased to 62% yield (entry 9). Higher yield (52%) could also be obtained using THF as solvent (entry 10). Further optimization of the reaction system on the basis of entry 9 and entry 10, we used NEt$_3$ as an additive, PhMe/THF (v/v 10:1) as solvent and this diboration reaction could proceed smoothly with lower catalyst (5%) and ligand loadings (5%) in 1 h, giving 2 in 78% yield (entry 11). The 1,1-diboration could not be proceeded in the absence of Ni catalyst (entry 12). Some other nickel sources were also tested, but they were not effective (entry 13, 14 and Supplementary Table 4).

**Scope of the methodology.** The substrate scope for the 1,1-diboration of aliphatic alkenes was shown in Table 2. A variety of nonactivated terminal aliphatic alkenes could be readily converted to the desired products with modest to high yields (25–78%). Substrates with different chain lengths (2–7)

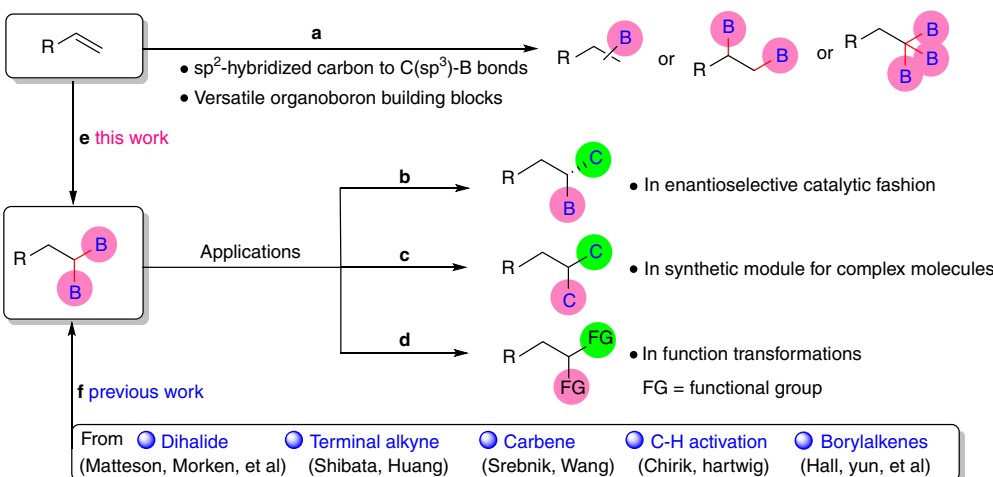

**Fig. 1** Borations of alkenes, synthetic methods and applications of 1,1-diborylalkanes. Transition metals catalyzed synthesis of monoboration (e.g., Rh[23], Fe[24–27], Co[28, 29], Ir[30], Ru[31]), 1,2-diboration (e.g., Rh[32, 33], Pt[34–36], Cu[37], Ni[38], Pd[39] and metal-free[40, 41]) and 1,1,1-triboration[10] of alkenes (**a**). Construction of chiral molecules utilizing racemic or nonracemic 1,1-diborylalkanes[11–15, 42, 43] (**b**). Conversion C–B bonds to C–C bonds to afford complex molecules (**c**). Transformations of organoboron building blocks into a wide variety of functional groups[44, 45] (**d**). Transition metal catalyzed synthesis of 1,1-diborylalkanes from terminal alkenes (**e**). Previous work was reported for preparation of 1,1-diborylalkanes from 1,1-dihalides[11, 21, 46], terminal alkynes[47–49], carbene insertion[50, 51], hydroboration of borylalkenes[42, 43] or C–H activation[52–54] (**f**)

**Table 1 Screening of conditions for the preparation of 1,1-diborylalkanes from alkenes**

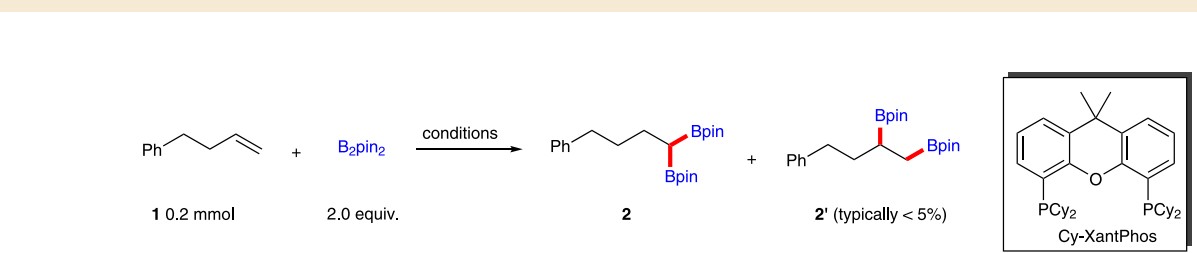

| Entry[a] | Ni cat. | Ligand | Base | Solvent | Yield (%)[b] |
|---|---|---|---|---|---|
| 1 | Ni(COD)$_2$ | PCy$_3$ | LiOMe | PhMe | 39 |
| 2 | Ni(COD)$_2$ | PCy$_3$ | LiOBu$^t$ | PhMe | <5 |
| 3 | Ni(COD)$_2$ | PCy$_3$ | Cs$_2$CO$_3$ | PhMe | 22 |
| 4 | Ni(COD)$_2$ | PCy$_3$ | NaOBu$^t$ | PhMe | Trace |
| 5 | Ni(COD)$_2$ | PCy$_3$ | KOBu$^t$ | PhMe | <5 |
| 6 | Ni(COD)$_2$ | PCpent$_3$ | LiOMe | PhMe | Trace |
| 7 | Ni(COD)$_2$ | PCyPh$_2$ | LiOMe | PhMe | Trace |
| 8 | Ni(COD)$_2$ | IMes·HCl | LiOMe | PhMe | trace |
| 9 | Ni(COD)$_2$ | Cy-XantPhos | LiOMe | PhMe | 62 |
| 10 | Ni(COD)$_2$ | Cy-XantPhos | LiOMe | THF | 52 |
| 11[c] | Ni(COD)$_2$ | Cy-XantPhos | LiOMe/NEt$_3$ | PhMe/THF | **78** |
| 12 | – | Cy-XantPhos | LiOMe/NEt$_3$ | PhMe/THF | 0 |
| 13 | Ni(PPh$_3$)$_4$ | PCy$_3$ | LiOMe | PhMe | 0 |
| 14 | NiCl$_2$(PCy$_3$)$_2$ | PCy$_3$ | LiOMe | PhMe | 12 |

Standard reaction conditions: 5% Ni(COD)$_2$, 5% Cy-XantPhos, 2.0 equiv. B$_2$pin$_2$, 1.0 equiv. LiOMe, 0.5 equiv. NEt$_3$, 0.55 mL PhMe/THF(v:v/10:1) with Ar protection at 130 °C for 1 h
[a]For entry 1–10 and 13–14, 10% Ni(COD)$_2$ and 20% Ligand, at 130 °C for 12 h
[b]GC yield average of two runs using $n$-tetracosane as internal standard
[c]Standard reaction conditions

could react smoothly. A host of different ethers, such as aryl ether (**8**–**10**), silyl ether (**12**) and benzyl ether (**20**) could participate well in the target reaction. Notably, methoxyl group amenable for Ni-catalyzed C-O cleavage did not compete with the efficacy of our 1,1-diboration event. As an important class of biologically active molecules, various nitrogen-containing moieties, such as indole (**13**), carbazole (**14**), and methyl benzylamine (**16**), amide (**16**, **17**), all survived the 1,1-diboration. Interestingly, multiboron compounds (**18**, **19**) could be smoothly synthesized from alkenes bearing Bpin moieties, thus providing an additional handle for further group transformations, cross-coupling reactions, or both. Furthermore, *R*-glyceraldehyde-acetonide derivative can be transformed into **20** in 78% yield. The diboration process is not affected by sterical hindrance of groups in mono-substituted alkenes (e.g., **21**). Internal alkenes could be well accommodated in this reaction, albeit achieving **22** and **23** only in acceptable yield. In line with the expected reaction result, 1,1-disubstituted alkene **25** and internal alkene (e.g., **22**) are ineffective substrates. Besides, B$_2$pai$_2$ can also be used as boron source and the desired product (**24**) was obtained in 56% yield. Underscoring the utility of our protocol, gram-scale diboration reactions of substrate 1-decene (5.61 g, 40.0 mmol) and 1-(hex-5-en-1-yl)-1*H*-indole (1.00 g, 5.0 mmol) were conducted under standard conditions respectively. Gratifyingly, there was very little lose in isolated yield of **6** (55%) and **13** (66%) (Table 2 and Supplementary Methods).

Next, we turned our attention to validate the generality of our 1,1-diboration protocol for vinylarenes (Table 3). To our disappointment, the optimized reaction conditions for terminal aliphatic alkenes have little effect on vinylarenes. However, fine-tuning of ligands on the basis of the reaction parameters, we found that tricyclohexyl phosphine could afford the desired product in satisfactory yields. Notably, similar results were obtained regardless of the electron-donating (e.g., **27**, **29**, **30**) or electron-withdrawing (**32**, **33**) groups on the substrates. Steric hindrance (e.g., **30**) shows subtle influence on the yield of the desired products.

**1,1-Diboration of sugar derivatives, LCs and lower alkenes**. This 1,1-diboration reaction of terminal alkenes was utilized for the modification of sugar derivatives, liquid crystals (**LCs**) and even ethylene shown in Fig. 2. *C*-alkyl or *O*-alkyl glycosides are important bioactive candidates[58]. It would therefore be interesting to use our reaction to modify glycosides to form 1,1-diboron-containing bioactive synthons. *C*-allyl D-glucose derivative **35** was synthesized from tetrabenzyl-protected D-glucose **34** via fluorination and subsequent allylation. 1,1-Diboration of **35** under the optimal reaction conditions, the desired product **36** was obtained in 63% yield. Furthermore, the *O*-alkyl substrate **38** can be transformed into **39** in 69% yield with the protecting group acetonide well tolerated, which is a frequently used protecting group in sugar chemistry. Fluorinated Liquid crystals (LCs), because of their high birefringence and small losses in terahertzband and susceptibility for electric and magnetic field, are excellent materials to construct tunable optical devices (e.g., phase shifters, phase gratings lenses, filters and metamaterials)[59]. Commercial available LCs **40** possessing a side alkenyl chain was successfully modified using our protocol. The desired product **41** was obtained in 47% yield. The two newly formed C-B bonds made **41** possible to transform in a more enriched way to afford other possible LCs molecules via further C–B bond transformations. We notice that double bond in the side carbon chain of LCs is not a necessary functional group, and many other useful LCs molecules could be derived from this double bond[59]. To further highlight the practicality of our protocol in lower alkenes, propylene **42** was utilized and it could be successfully 1,1-diborated.

**Table 2 Substrate scope for 1,1-diboration of aliphatic alkenes**[a]

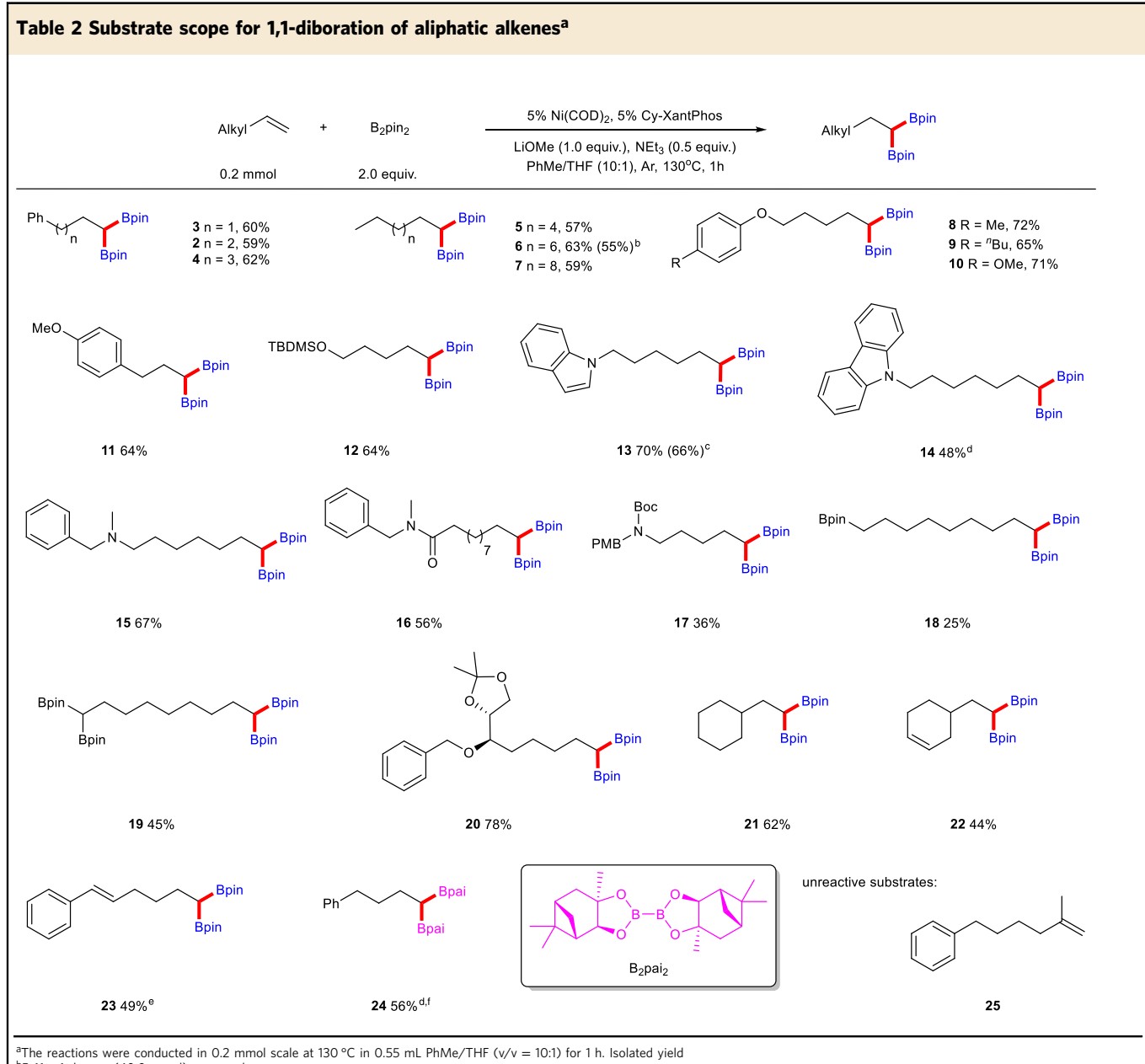

[a]The reactions were conducted in 0.2 mmol scale at 130 °C in 0.55 mL PhMe/THF (v/v = 10:1) for 1 h. Isolated yield
[b]5.61 g 1-decene (40.0 mmol) was used
[c]1.00 g 1-(hex-5-en-1-yl)-1H-indole (5.0 mmol) was used. Isolated yield of gram-scale is shown in a parenthesis
[d]10% Ni(COD)₂ and 10% Cy-XantPhos, 4h
[e]Yield was determined by 1H NMR using diphenyl methane as internal standard
[f]1.1 mL PhMe/THF (v/v = 10:1). TBDMS = dimethyl-tert-butylsilyl. Boc = t-butyloxy carbonyl. PMP = p-methoxybenzyl. B₂pai₂ = bis[(+)-pinanediolato]diboron

The desired product **43** was isolated in 82% yield. Performing diboration of another kind of lower alkenes ethylene **44** afforded the 1,1-diboration product **45** and 1,2-diboration undesired product **46** in 45% combined yield, and the ratio of **45/46** was 1:2 (Supplementary Fig. 56). Diboration results of **42** and **44** indicate that steric hindrance in alkenes plays a positive role in this site-selective matter. The above transformations demonstrated a vast application prospect of our developed reaction.

## Discussion

A set of experiments were executed as additional evidence in support of our proposed catalytic cycle (Supplementary Discussions). The cross-over experiment of **1** with B₂pin₂ and B₂pai₂ to yield **2**, **47** and **24** indicates that the two boron motifs in

the products were supplied by two molecular bisboronic ester (Fig. 3a). The result of cross-over experiment of **1** with B₂pin₂ and B₂pin₂-$d_{12}$ is also in accordance with the above observations (Supplementary Discussions and Supplementary Fig. 61 for more details). Deuterium labeling studies were conducted by using 4-(2,2-dideuterovinyl)-1,1′-biphenyl (**29a**-$d_2$) as the substrate (93% deuterium content) and one of benzylic hydrogen atoms in the product was 92% deuterated (Fig. 3b). This deuterium experiment illustrated that one of the benzylic hydrogen atoms is originated from hydrogen atoms bonded in terminal site of alkenes. Through nickel-catalyzed alkenes boration and subsequent hydroboration process, we realize the 1,1-selective diboration of alkenes. Diboration of alkenes was all proceeded in a 1,2-selective manner before this study.

In summary, we have successfully developed a *chemo-* and *regio*-selective Ni-catalyzed 1,1-diboration reaction of terminal alkenes. Tremendous variety of terminal alkenes (vinylarenes, aliphatic alkenes and low alkenes) could react smoothly in this methodology. This reaction provides an effective and convenient way for the synthesis of 1,1-diborylalkanes. The obtained products provides an excellent platform for achieving complex structures and via C–C bond formation, function

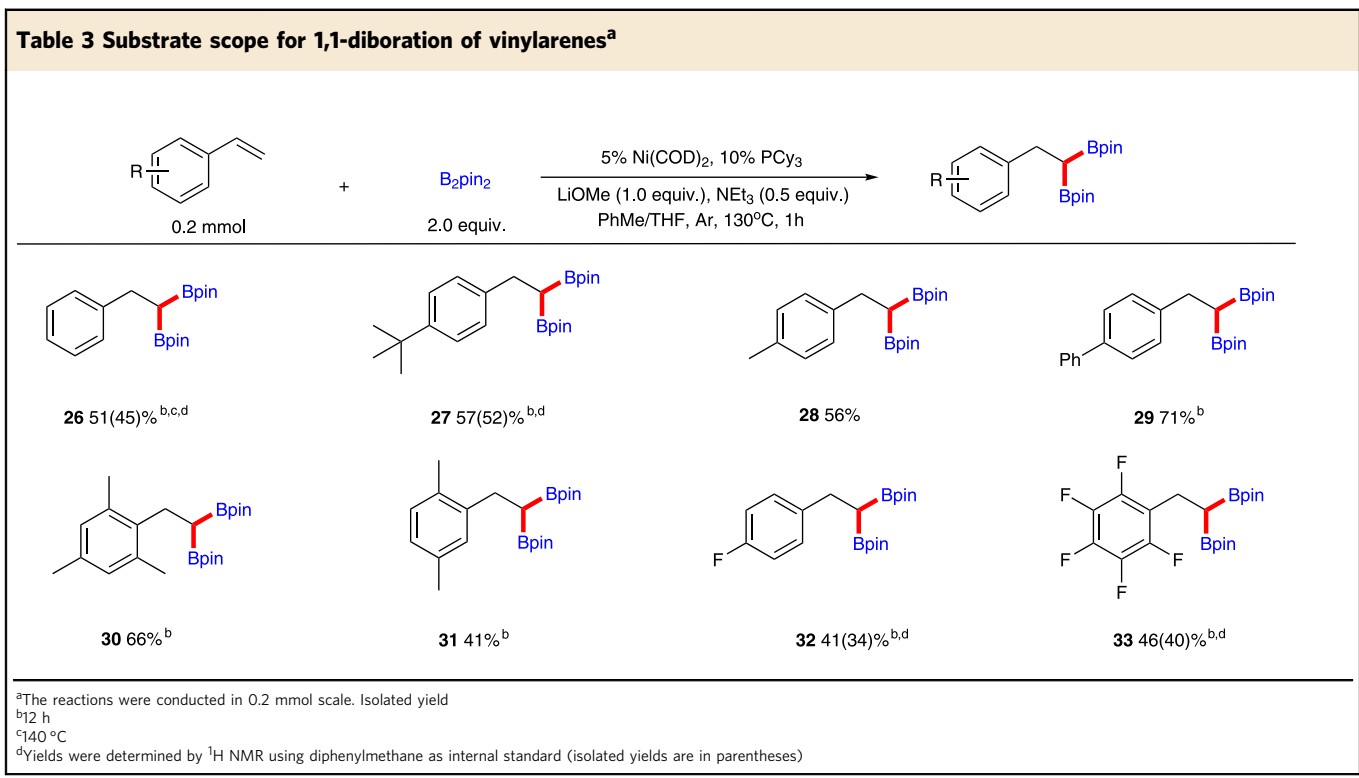

**Table 3 Substrate scope for 1,1-diboration of vinylarenes[a]**

**26** 51(45)%[b,c,d]    **27** 57(52)%[b,d]    **28** 56%    **29** 71%[b]

**30** 66%[b]    **31** 41%[b]    **32** 41(34)%[b,d]    **33** 46(40)%[b,d]

[a]The reactions were conducted in 0.2 mmol scale. Isolated yield
[b]12 h
[c]140 °C
[d]Yields were determined by [1]H NMR using diphenylmethane as internal standard (isolated yields are in parentheses)

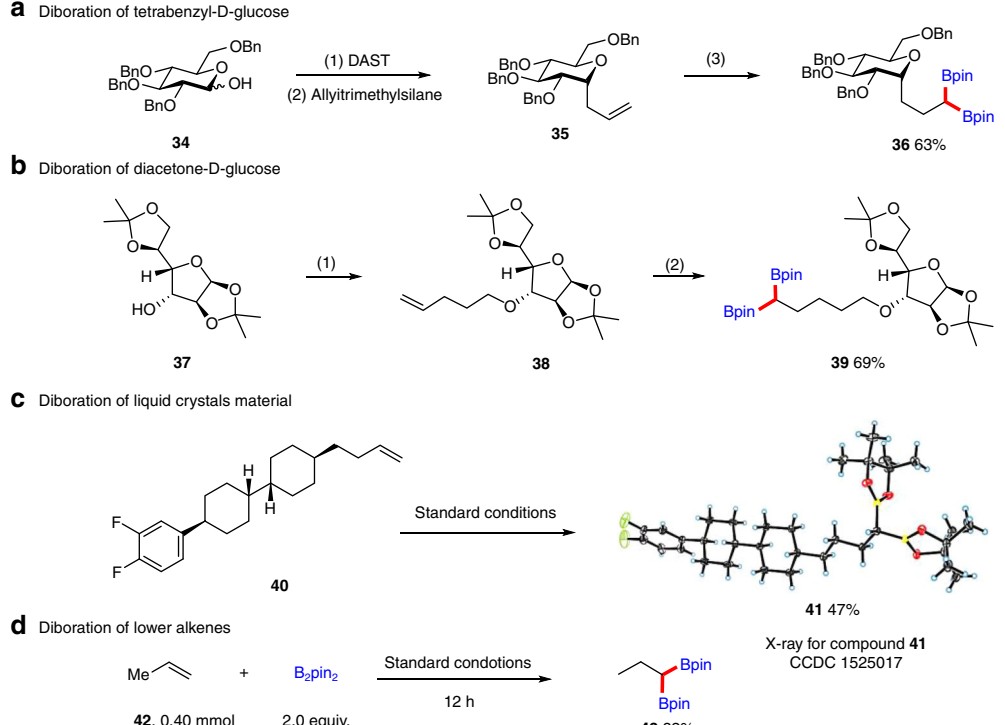

**a** Diboration of tetrabenzyl-D-glucose

**34** → (1) DAST, (2) Allyltrimethylsilane → **35** → (3) → **36** 63%

**b** Diboration of diacetone-D-glucose

**37** → (1) → **38** → (2) → **39** 69%

**c** Diboration of liquid crystals material

**40** → Standard conditions → **41** 47%

X-ray for compound **41**
CCDC 1525017

**d** Diboration of lower alkenes

Me + B₂pin₂ → Standard condotions, 12 h → **43** 82%

**42**, 0.40 mmol    2.0 equiv.

**Fig. 2** 1,1-Diboration of sugar derivatives, LCs and lower alkenes. **a** (1) DAST (diethylaminotrifluorosulfur, 1.1 equiv.), THF, −30 °C. (2) Allyltrimetnylsilane (2.0 equiv.), 20% BF₃ · Et₂O, DCM. (3) 10% Ni(COD)₂, 10% Cy-Xantphos. **b** (1) NaH, DMF, 5-bromopent-1-ene, 0 °C. (2) Standard conditions, 4 h. **c** Standard conditions. **d** Standard conditions, 12 h

**Fig. 3** Mechanism study experiments. **a** Cross-over experiments of **1** with B$_2$pin$_2$ and B$_2$pai$_2$ to afford 1,1-diboron compounds **2**, **47** and **24** respectively. GC peak area ratio of **2**, **47**, **24** is 1/2.2/1.1 (Correction factor is not taken into account). **b** Deuterium labeling experiment was conducted using **29a**-$d_2$ as starting material

transformations and even enantioselective catalytic cross-coupling reactions. Novel reactions of 1,1-diborylalkanes are being explored in our laboratory.

## Methods

**General procedure for 1,1-diboration of aliphatic alkenes**. In a glove box, a mixture of 5% Ni(COD)$_2$ (2.8 mg, 0.01 mmol), 5% Cy-XantPhos (6.0 mg, 0.01 mmol), LiOMe (7.6 mg, 0.2 mmol, 1.0 equiv.) and B$_2$pin$_2$ (101.6 mg, 0.4 mmol, 2.0 equiv.) were added to a Schlenk tube equipped with a stir bar. The vessel was evacuated and filled with argon for three cycles. 0.55 mL PhMe/THF (v/v, 10:1), NEt$_3$ (14 μL, 0.5 equiv.) and alkenes (0.2 mmol, 1.0 equiv.) were added respectively under a positive flow of argon. The reaction mixture was stirred at room temperature (r.t.) for seconds and then transferred to a pre-heated 130 °C oil bath. After 1 h stirring, the reaction mixture was cooled to r.t. and then diluted with DCM, filtered through silica gel with copious washings by EtOAc, concentrated, and purified by silica gel chromatography to afford the desired product.

**Data availability**. The authors declare that the data supporting the findings of this study are available within the article and its Supplementary Information files. For the experimental procedures, see Supplementary Methods. For NMR, GC-MS and HRMS analysis of the compounds in this article, see Supplementary Figs 1–62. For X-ray data of compound **41** (CCDC No. 1525017), see Supplementary Table 5 and Supplementary Fig. 63. The crystal data can be obtained free of charge from the Cambridge Crystallographic Data Center (www.ccdc.cam.ac.uk).

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

## Acknowledgements

We thank the support from the National Natural Science Foundation of China (21325208, 21472181, 21572212 and 21502184), Ministry of Science and Technology of China (2017YFA0303500), Chinese Academy of Science (XDB20000000), Youth Innovation Promotion Association of the Chinese Academy of Sciences (2015371), FRFCU and PCSIRT.

## Author contributions

L.L. designed and carried out the experimental work. X.L. and T.G. helped to complete the experimental work. B.X. and Y.F. directed the project and wrote the manuscript.

## Additional information

**Competing interests:** The authors declare no competing financial interests

