## [Peer Review File · Nature Communications]

Reviewers' comments:

Reviewer #1 (Remarks to the Author):

Within this manuscript, Xiao and Fu describe the synthesis of gem-diborylalkanes from 1-alkenes using nickel-catalyzed alkene dehydrogenative borylation and hydroboration. This work represents the first example of the preparation of such synthetically valuable compounds with readily accessible terminal alkenes as the starting materials. The method works for both aromatic alkenes and aliphatic alkenes. The reaction with the latter is of particular interest because the dehydrogenative borylation of aliphatic alkenes is often more challenging than that with aromatic alkenes. Although the yields of 1,1-diborylalkane products in general are moderate to good, the present method provides a convenient approach to access gem-diborylalkanes from simple starting materials. With that being said, I support the publication of this manuscript after addressing the following points.

- 1) It is necessary to add the other synthetic routes to 1,1-diborylalkanes, for examples, the dual hydroboration of alkynes, the carbene transfer reactions by Jianbo Wang, and the direct double borylation by Hartwig. These could be added in Figure 1 and a concise introduction should be included accordingly.
- 2) Did the authors observe other boron-containing products, such as the monoborylalkanes or 1,2-diborylalkanes, in table 2 and table 3?
- 3) I do not know how to address it, but the current description, "modification of important molecules" seems to be odd. The definition of "important molecules" could vary. To some readers, some molecules shown in tables 2 and 3 can be more important than those in Figure 2.
- 4) A quantitative ¹³C NMR analysis of H/D scrambling is appropriate for the deuterium labeling experiment.
- 5) Adding a mechanistic cycle will be useful to the readers.
- 6) In figure 1, one carbon atom is missing in the diborylalkane product.

Reviewer #2 (Remarks to the Author):

The paper by Fu and co-workers reports a Ni-catalyzed synthesis of 1,1-diborylalkanes from terminal alkenes. The reaction employs 5 mol % of commercially available nickel salt and a phosphine ligand, in combination with an alkoxide base to deliver products efficiently and in a site-selective manner. The authors demonstrate good scope across broad substrate range. In particular, the reaction is tolerant of terminal alkenes (aryl and alkyl) bearing various functional groups such as ethers, amides, amines, and alkenes. Tolerance of the reaction to more complex as well as very simple substrates is highlighted through the catalytic diborylation of terminal alkenes within sugars, a liquid crystal, and ethylene. The authors also provide mechanistic experiments that show that two molecules of B₂(pin)₂ are required, and that deuterium scrambling takes place. The Supporting information adequately supports the claims in the paper, barring a few errors (see below).

While 1,1-diborylalkanes are emerging useful building blocks for chemical synthesis, many 1,1-diborylalkanes can already be synthesized by alternative stoichiometric and catalytic methods. Such methods include: (1) alkylation of diborylmethane (e.g., Matteson, Morken, Meek, Shibata, etc.); (2) hydroboration of 1-alkenyl organoborons (Yun, *Angew. Chem. Int. Ed.* 2013, 52, 1); (3) isomerization/hydroboration (Chirik, *Org. Lett.*, 2015, 17, 2716), (4) C–H functionalization (Chirik, *JACS*, 2016, 138, 766); 1,1-dihalide substitution (Morken, *JACS*, 2014, 136, 10581); (5) Diboration of a terminal alkene (not very efficient, Murakami, *Angew. Chem. Int. Ed.* 2015, 54, 12659). In the submitted manuscript by Fu and co-workers, the catalytic synthesis of 1,1-diborylalkanes from terminal alkenes is clearly a significant advance in this area, however, it will not particularly alter the thinking of how people in the field utilize 1,1-diborylalkanes. For example, the reported method by Fu requires a terminal alkene, and as such cannot be used to synthesize

the 1,1-diborylalkanes made by C-H functionalization or alkylation (see refs above). Hence, the methods are complementary, and one is not necessarily superior over the other. The paper also leaves a number of questions unanswered. For example: (1) What does being able to form a 1,1-diboron from an alkene of a liquid crystal and sugars enable. Granted, they are more complicated substrates and interesting reactions but fall short of showing why the molecules efficiently made this way are significant. (2) Could the ethylene result be improved? The diboration of ethylene serves as a very nice and useful way to make 1,1-diborylethane, however, the reaction seems quite underwhelming. The borylation of ethylene produces a 42% combined yield (1:2.5 selectivity), which corresponds to a 12% yield of desired product.

In general, the reported catalytic method will impact the efficiency with which synthetic chemists prepare 1,1-diborylalkanes, but falls short in showcasing how 1,1-diborylalkanes synthesized by this approach prove advantageous in chemical synthesis. Overall, this is a nice paper describing practical catalytic method that will be of interest to the synthetic community but not necessarily the wider field, and unfortunately does not meet the level of importance to be published in Nature.

Below is a list of additional comments/suggestions/errors that should be taken into consideration by the authors.

- 1) SI Missing silica gel chromatography conditions for 1,1-diborylalkane products.
- 2) SI missing physical appearance of product (I.e., colorless oil, white solid, etc.)
- 3) SI missing, ¹H NMR yields, isolated yields, and mass data for vinyl arenes. Only the spectral data is included. The missing information should be added.
- 4) Products only have three pieces of characterization data, where relevant IR for key functional groups (e.g., amides carbonyl), should be included.
- 5) All calculated HRMS values are 0.0004–0.0006 greater than what they should be. This is consistent throughout SI and seems like a simple error, which can easily be corrected.
- 6) In some instances, characterization data for the alkenes substrates is given but the synthetic procedures to make them is absent, unless a references are missing the experimental procedures used should be included.
- 7) Optical rotations are missing for chiral compounds 20, 24, 36, and 39.
- 8) Compound 41 has multiple signals in the ¹⁹F NMR likely due to chair conformations or diastereoisomers? A comment as to why there are some many signals should be added.
- 9) In the manuscript, all alkenes have a wedge, this should be removed.
- 10) 41 is an achiral compound and its name in the SI should not contain (R) stereochemical descriptor.
- 11) A proposed mechanism would be very helpful based on the mechanistic data provided.
- 12) For such a practical process, a gram-scale reaction to highlight practicality is noticeably absent from the paper.

To Reviewer #1:

“Within this manuscript, Xiao and Fu describe the synthesis of gem-diborylalkanes from 1-alkenes using nickel-catalyzed alkene dehydrogenative borylation and hydroboration. This work represents the first example of the preparation of such synthetically valuable compounds with readily accessible terminal alkenes as the starting materials. The method works for both aromatic alkenes and aliphatic alkenes. The reaction with the latter is of particular interest because the dehydrogenative borylation of aliphatic alkenes is often more challenging than that with aromatic alkenes. Although the yields of 1,1-diborylalkane products in general are moderate to good, the present method provides a convenient approach to access gem-diborylalkanes from simple starting materials. With that being said, I support the publication of this manuscript after addressing the following points.”

Response: We thank **reviewer #1** for the positive comments and supporting the publication of this manuscript.

“(1) It is necessary to add the other synthetic routes to 1,1-diborylalkanes, for examples, the dual hydroboration of alkynes, the carbene transfer reactions by Jianbo Wang, and the direct double borylation by Hartwig. These could be added in Figure 1 and a concise introduction should be included accordingly.”

Response: Synthetic routes to 1,1-diborylalkanes (Fig. 1 f) and related concise introduction were added in revised manuscript. We added references 46-54 in the revised manuscript.

“(2) Did the authors observe other boron-containing products, such as the monoborylalkanes or 1,2-diborylalkanes, in table 2 and table 3?”

Response: We observe less than 5% terminal monoborylalkene and trace amount of 1,2-diborylalkane. Though some 1,1-diboration reactions showed moderate yields, we are definitely sure that the 1,1-diboration products are absolutely main products in table 2 and table 3.

“(3) I do not know how to address it, but the current description, “modification of important molecules” seems to be odd. The definition of “important molecules” could vary. To some readers, some molecules shown in tables 2 and 3 can be more important than those in Figure 2”

Response: The description “modification of important molecules” was adjusted to “1,1-diboration of sugar derivatives, LCs and lower alkenes” in the revised manuscript.

“(4) A quantitative ¹³C NMR analysis of H/D scrambling is appropriate for the deuterium labeling experiment.”

Response: A quantitative ¹³C NMR analysis of compound **29-d₂** was measured (4s and 8s relaxation delay, 1024 scans). The peaks from benzylic and homobenzylic carbons should be split into triplet. However, we were failed to observe this splitting, because the signals were very weak. And we checked ¹H NMR and HRMS data carefully to make sure the correctness of deuterium atom migration.

“(5) Adding a mechanistic cycle will be useful to the readers.”

Response: We conducted a set of mechanism experiments (e.g., crossover experiments and deuterium labelling experiments) and achieved some key mechanistic information (e.g., the origin of two boron motifs in products and the migration hydrogen atom in alkenes) (see SI, P65-P69). A relatively probable catalytic cycle was proposed based on our obtained information and we added the cycle in SI.

“(6) In Figure 1, one carbon atom is missing in the diborylalkane product.”

Response: We corrected this typo and checked the manuscript carefully to avoid alike typos.

To Reviewer #2:

*“The paper by Fu and co-workers reports a Ni-catalyzed synthesis 1,1-diborylalkanes from terminal alkenes. The reaction employs 5 mol % of commercially available nickel salt and a phosphine ligand, in combination with an alkoxide base to deliver products efficiently and in a site-selective manner. The authors demonstrate good scope across broad substrate range. In particular, the reaction is tolerant of terminal alkenes (aryl and alkyl) bearing various functional groups such as ethers, amides, amines, and alkenes. Tolerance of the reaction to more complex as well as very simple substrates is highlighted through the catalytic diborylation of terminal alkenes within sugars, a liquid crystal, and ethylene. The authors also provide mechanistic experiments that show that two molecules of $B_2(pin)_2$ are required, and that deuterium scrambling takes place. The Supporting information adequately supports the claims in the paper, barring a few errors (see below). While 1,1-diborylalkanes are emerging useful building blocks for chemical synthesis, many 1,1-diborylalkanes can already be synthesized by alternative stoichiometric and catalytic methods. Such methods include: (1) alkylation of diborylmethane (e.g., Matteson, Morken, Meek, Shibata, etc.); (2) Hydroboration of 1-alkenyl organoborons (Yun, *Angew. Chem. Int. Ed.* 2013, **52**, 1); (3) isomerization/hydroboration (Chirik, *Org. Lett.*, 2015, **17**, 2716); (4) C–H functionalization (Chirik, *JACS*, 2016, **138**, 766). 1,1-dihalide substitution (Morken, *JACS*, 2014, **136**, 10581); (5) Diboration of a terminal alkene (not very efficient, Murakami, *Angew. Chem. Int. Ed.* 2015, **54**, 12659). In the submitted manuscript by Fu and co-workers, the catalytic synthesis of 1,1-diborylalkanes from terminal alkenes is clearly a significant advance in this area, however, it will not particularly alter the thinking of how people in the field utilize 1,1-diborylalkanes. For example, the reported method by Fu requires a terminal alkene, and as such cannot be used to synthesize the 1,1-diborylalkanes made by C-H functionalization or alkylation (see refs above). Hence, the methods are, and one is not necessarily superior over the other.”*

Response: The emerging utility of 1,1-diborylalkanes has been demonstrated in chemical synthesis. Substantial progress has been made in the development of C-C bond forming reactions. Plenty of pioneering work has been reported by Morken, Meek, Shibata and other groups (The relevant work was cited in ref. 11-17, 19-22. We also added ref. 18 in revised manuscript). Considering question 1 raised by **reviewer #1** and previous synthetic methods of 1,1-diborylalkanes mentioned by **reviewer #2**, we added previous synthetic routes to 1,1-diborylalkanes in revised manuscript (Fig. 1 f), so the caption of old Fig. 1 was rewritten. We admit that some 1,1-diborylalkanes (e.g., 1,1-dialkyldiborylalkanes) could not be synthesized using our protocol. The

above mentioned synthetic methods possess their own superiorities as well, our synthetic method shows different reaction results (e.g., eq. 1, 2) and is more superior to others in the synthesis of some diboron compounds (e.g., compounds **36** and **41**). Moreover, it was characterized by its ready availability of diverse alkenes (vinylarenes, aliphatic alkenes and lower alkenes), conversion of which is always the research focus in our lab (e.g., *Nat. Commun.*, 2016, **7**, 11129.; *Angew. Chem. Int. Ed.*, 2015, **54**, 12957.).

“The paper also leaves a number of questions unanswered. For example: (1) What does being able to form a 1,1-diboron from an alkene of a liquid crystal and sugars enable. Granted, they are more complicated substrates and interesting reactions but fall short of showing why the molecules efficiently made this way are significant.”

Response: The two newly formed C-B bonds made **41** possible to transform in a more enriched way to afford other possible LCs molecules *via* further C-B bond transformations. We noticed that double bond in the side carbon chain of liquid crystals is not a necessary functional group, and derivation on this double bond can derive many other useful LCs molecules. (*Crystals*, 2013, **3**, 443. P467, table 14; ref. 59 Chart 1). The same diverse conversions could also be conducted on **36** and **39**. We added above description in revised manuscript. If **reviewer #2** thinks these diborylated molecules are not significant enough to be listed as a separate scheme, we are agreed to move these substrates to table 2.

“(2) Could the ethylene result be improved? The diboration of ethylene serves as a very nice and useful way to make 1,1-diborylethane, however, the reaction seems quite underwhelming. The borylation of ethylene produces a 42% combined yield (1:2.5 selectivity), which corresponds to a 12% yield of desired product.”

Response: Despite our best efforts (fine-tuning reaction parameters, using high pressure of ethylene), the selectivity was slightly improved to 1:2 with 45% combined yield (Details of optimization results were added in SI, P79). Fortunately, propylene, as one of the most important low alkenes, could be 1,1-diborylated to afford desired product in high yield (82%) (SI, P61). Diboration results of propylene and ethylene indicate that steric hindrance in alkenes plays a positive role in this site-selective matter. To better highlight practicability in low alkenes, propylene took the place of ethylene in revised manuscript (Fig. 2) and related work of ethylene was removed to SI.

“In general, the reported catalytic method will impact the efficiency with which synthetic chemists prepare 1,1-diborylalkanes, but falls short in showcasing how 1,1-diborylalkanes synthesized by this approach prove advantageous in chemical synthesis. Overall, this is a nice paper describing practical catalytic method that will be of interest to the synthetic community but not necessarily the wider field, and unfortunately does not meet the level of importance to be published in Nature.”

Response: We thank **reviewer #2** for the positive comments. Our protocol presents many advantages including extensive sources of alkenes, high *chem*- and *regio*-selectivity, good functional group compatibility and mild to high yields. The **reviewer #2** helpful comments and suggestions have significantly improved the quality of our manuscript. We hope the revised manuscript will be publishable in *Nature Communications*.

“Below is a list of additional comments/suggestions/errors that should be taken into consideration by the authors.

(1) SI Missing silica gel chromatography conditions for 1,1-diborylalkane products.

(2) SI missing physical appearance of product (i.e., colorless oil, white solid, etc.)

(3) SI missing, ¹H NMR yields, isolated yields, and mass data for vinyl arenes. Only the spectral data is included. The missing information should be added.

(4) Products only have three pieces of characterization data, where relevant IR for key functional groups (e.g., amides carbonyl), should be included.

(5) All calculated HRMS values are 0.0004–0.0006 greater than what they should be. This is consistent throughout SI and seems like a simple error, which can easily be corrected.

(6) In some instances, characterization data for the alkenes substrates is given but the synthetic procedures to make them is absent, unless references are missing the experimental procedures used should be included.

*(7) Optical rotations are missing for chiral compounds **20**, **24**, **36**, and **39**.*

*(8) Compound **41** has multiple signals in the ¹⁹F NMR likely due to chair conformations or diastereoisomers? A comment as to why there are some many signals should be added.*

(9) In the manuscript, all alkenes have a wedge, this should be removed.

*(10) **41** is an achiral compound and its name in the SI should not contain (R)-stereochemical descriptor.*

(11) A proposed mechanism would be very helpful based on the mechanistic data provided.

(12) For such a practical process, a gram-scale reaction to highlight practicality is noticeably absent from the paper.”

Response: All these were carefully revised in manuscript and SI.

(1): Silica gel chromatography conditions for substrates and products were added in SI (P71-P94).

(2): Physical appearance of substrates and products were added in SI (P71-P94).

(3): All isolated yields for vinylarene products (added in revised manuscript), relevant HRMS data was added in SI (P87-P90).

(4): IR spectra of substrates and products were added in SI (P71-P94).

(5): HRMS values were measured under positive ion mode, so all our calculated HRMS values of substrates and products are derived from the sum of exact mass of molecules and hydrogen (or sodium) positive ion. These calculated values are consistent with the calculated values measured by MALDI-TOF MS system from analysis center. In fact, the exact electronic mass (0.0004-0.0006) should not be summed in calculated values.

(6): Experimental procedures of substrates were added in SI (P73, 74, 92).

(7): Optical rotations of compounds **20**, **24**, **36**, **39** were measured and the values were added in SI (P85, 87, 91). Besides, the related starting materials were also measured and the values were added in SI (P75-P77).

(8): Fluoro-containing byproduct was collected as well when isolating compound **41**. Such is the fact when we further purified compound **41** and found there are two signals in the ^{19}F NMR. The ^{19}F NMR spectra and shift were revised in SI (P49, 92).

(9): The wedges in all alkenes were removed.

(10): The chemical name of compound **41** was corrected as 4-{*trans*-4-[*trans*-4-(3,4-difluorophenyl)cyclohexyl]cyclohexyl}butylbis(4,4,5,5-tetramethyl-1,3,2-dioxaborolane) in SI (P91).

(11): We conducted a set of mechanism experiments (e.g., crossover experiments and deuterium labelling experiments) and achieved some key mechanistic information (e.g., the origin of two boron motifs in products and the migration of terminal hydrogen atom in alkenes) (see SI, P65-P69). A relatively probable catalytic cycle was proposed based on our obtained information and we added the cycle in SI.

(12): Two gram-scale reactions were conducted.

Silica gel

REVIEWERS' COMMENTS:

Reviewer #1 (Remarks to the Author):

The revised manuscript by Fu and Xiao has addressed my concerns. A significant amount of new data has been added in comparison to the original version. I support its publication in Nature Communication.

Reviewer #2 (Remarks to the Author):

The revised manuscript, and updated experimental, by Xiao and Fu are significantly improved. The changes made by the authors now places the Ni-catalyzed protocol into better context with previously reported methods. Furthermore, improvements such as (1) the more efficient reaction propylene instead of ethylene, (2) addition of a gram-scale reaction, (3) a catalytic cycle, and (4) improvements to the SI and data, increase the overall impact of the paper.

One criticism that still remains unaddressed relates to functionalization of the previously described "important molecules" (Figure 2). Alkyl 1,1-diborons certainly are useful compounds for synthesis – why not show a functionalization? For example, while suggesting compounds 36, 39, or 41 can be modified due to the diverse reactions available to the C–B(pin) moiety, it would be useful to see what type of reactions the authors envisioned would be important for 36, 39, or 41 that are not accessible from the corresponding alkene.

Overall, the revised paper (and experimental) adequately addresses many of the concerns raised by this reviewer. While functionalization of the products is not showcased, in its updated form the Ni-catalyzed protocol represents a useful advance in the field that will be of interested to the broad readership of Nature Communications. I recommend accepting without revisions.

Errors/ Comments:

- (1) Page 7, line 179: The sentence begins with "Erenow"
- (2) Although it might not be possible due to manuscript length, having the proposed catalytic cycle in manuscript to go along with the mechanistic experiments in Figure 3 would be helpful to the reader, compared to than putting the scheme in the SI.

To Reviewer #1:

“The revised manuscript by Fu and Xiao has addressed my concerns. A significant amount of new data has been added in comparison to the original version. I support its publication in Nature Communication.”

Response: We thank **reviewer #1** for supporting the publication of our manuscript.

To Reviewer #2:

“The revised manuscript, and updated experimental, by Xiao and Fu are significantly improved. The changes made by the authors now place the Ni-catalyzed protocol into better context with previously reported methods. Furthermore, improvements such as (1) the more efficient reaction propylene instead of ethylene, (2) addition of a gram-scale reaction, (3) a catalytic cycle, and (4) improvements to the SI and data, increase the overall impact of the paper.”

Response: We thank **Reviewer #2** for the positive comments on improvements in revised manuscript.

“One criticism that still remains unaddressed relates to functionalization of the previously described “important molecules” (Figure 2). Alkyl 1,1-diborons certainly are useful compounds for synthesis – why not show a functionalization? For example, while suggesting compounds 36, 39, or 41 can be modified due to the diverse reactions available to the C–B(pin) moiety, it would be useful to see what type of reactions the authors envisioned would be important for 36, 39, or 41 that are not accessible from the corresponding alkene.”

Response: The related derivations of 1,1-diboron compounds (e.g. compound **36**) are undertaking in our lab and will be reported soon.

“Overall, the revised paper (and experimental) adequately addresses many of the concerns raised by this reviewer. While functionalization of the products is not showcased, in its updated form the Ni-catalyzed protocol represents a useful advance in the field that will be of interested to the broad readership of Nature Communications. I recommend accepting without revisions.”

Response: We thank **Reviewer #2** for the positive comments and supporting publication of our manuscript.

“Errors/ Comments:

(1) Page 7, line 179: The sentence begins with “Erenow”

(2) Although it might not be possible due to manuscript length, having the proposed catalytic cycle in manuscript to go along with the mechanistic experiments in Figure 3 would be helpful to the reader, compared to than putting the scheme in the SI.”

Response: (1) The sentence that begins with “Erenow” was rewrote as “Diboration of alkenes was all proceeded in a 1,2-selective manner before this study”.

(2) Considering the manuscript length and mechanistic rigor of this paper, putting the catalytic cycle in SI would be better.